# Closed-Loop Control of Droplet Transfer in Electron-Beam Freeform Fabrication

**DOI:** 10.3390/s20030923

**Published:** 2020-02-10

**Authors:** Shuhe Chang, Haoyu Zhang, Haiying Xu, Xinghua Sang, Li Wang, Dong Du, Baohua Chang

**Affiliations:** 1Department of Mechanical Engineering, Tsinghua University, Beijing 100084, China; changsh15@mails.tsinghua.edu.cn (S.C.); z-hy18@mails.tsinghua.edu.cn (H.Z.); wanglidme@mail.tsinghua.edu.cn (L.W.); 2Key Laboratory for Advanced Materials Processing Technology, Ministry of Education, Tsinghua University, Beijing 100084, China; 3Manufacturing Technology Institute, Aviation Industry Corporation of China, Beijing 100024, China; xhyxhy@126.com (H.X.); sang_xh@163.com (X.S.)

**Keywords:** metal additive manufacturing, electron-beam freeform fabrication, droplet transfer, closed-loop control

## Abstract

In the process of electron-beam freeform fabrication deposition, the surface of the deposit layer becomes rough because of the instability of the feeding wire and the changing of the thermal diffusion condition. This will make the droplet transfer distance change in the deposition process, and the droplet transfer cannot always be stable in the liquid bridge transfer state. It is easy to form a large droplet or make wire and substrate stick together, which makes the deposition quality worsen or even interrupts the deposition process. The current electron-beam freeform fabrication deposition is mostly open-loop control, so it is urgent to realize the real-time and closed-loop control of the droplet transfer and to make it stable in the liquid bridge transfer state. In this paper, a real-time monitoring method based on machine vision is proposed for the droplet transfer of electron-beam freeform fabrication. The detection accuracy is up to ± 0.08 mm. Based on this method, the measured droplet transfer distance is fed back to the platform control system in real time. This closed-loop control system can stabilize the droplet transfer distance within ± 0.14 mm. In order to improve the detection stability of the whole system, a droplet transfer detection algorithm suitable for this scenario has been written, which improves the adaptability of the droplet transfer distance detection method by means of dilatation/erosion, local minimum value suppression, and image segmentation. This algorithm can resist multiple disturbances, such as spatter, large droplet occlusion and so on.

## 1. Introduction

In recent years, the technology of metal additive manufacturing has been developed rapidly, attracting the attention of many researchers. There are many ways to additively manufacture metals, including Electron Beam Melting (EBM), Selective Laser Melting (SLM), Direct Metal Laser Sintering (DMLS), Hot Wire Gas Tungsten Arc Welding (HW-GTAW), and Ion Fusion Formation (IFF). The most remarkable feature of additive manufacturing is that a product is made layer-by-layer without mold or tooling. The computer aided design (CAD) model is divided into layers of consistent thickness by slicing software. Only one layer is deposited at a time, and the layers are stacked layer-by-layer to form the final part [1,2]. Electron-Beam Freeform Fabrication (EBF3) is one of the metal additive manufacturing technologies. The processed parts can reach the strength of forgings. Compared with other metal additive manufacturing methods, EBF3 can achieve a higher deposition efficiency with quality assurance because the vacuum environment is beneficial to the protection of parts [3,4].

The instability of the droplet transfer will, however, cause the quality of the electron-beam freeform fabrication to worsen or even cause the interruption of the deposition process [5,6,7]. It is urgent to realize the closed-loop control of the droplet transfer, which will make the deposition process stable in the liquid bridge transfer state, but the rough surface of the deposited layer will cause instability of the droplet transfer [8,9,10,11,12]. Geometric errors occur because of the instability of the heat input and change in the thermal diffusion conditions. The surface of the deposition layer will become obviously uneven after layer-by-layer accumulation, even if the geometric error is very small in each layer. The height of the wire feeder is usually fixed during the EBF3, so when the surface of the deposited layer is higher than the target surface, the droplet transfer distance decreases, and the wire and substrate become easier to adhere to one another, resulting in the interruption of the deposition process. Further, when the surface of the deposited layer is concave, the droplet transfer distance increases, which results in a large droplet at this time and seriously affects the forming quality [5]. In order to ensure the deposition quality, it is urgent to control the droplet transfer distance in real time so that the deposition process is stable in the transfer state of the liquid bridge. In order for electron-beam freeform fabrication technology to be applied in the environment of space, a stable liquid bridge transfer must also be realized. When deposited on Earth, the droplets are still able to transfer by gravity. In the environment of space, however, weightlessness makes it impossible to rely on gravity for droplet transfer—only surface tension can be relied on for droplet transfer [13]. This requires the deposition process to maintain a very stable liquid bridge transfer.

Optical images are often used for real-time closed-loop control because of their adequate information. Heralić et al. [14,15] detected the height of the deposition layer offline using optical methods, but could not achieve real-time control. Zeng et al. [16,17,18] proposed a welding pass-detection method based on directional light and structure-light information fusion, aiming to tackle the problem that the structure-light laser could not obtain the welding pass information stably under the condition of strong mirror reflection. However, in the process of EBF3, the lens in front of the laser will be coated with a layer of metal vapor which makes the light transmittance drop sharply and, therefore, not meet the need of long-time stable work. Chang et al. [19,20] realized the 3D reconstruction of the electron-beam freeform fabrication process based on electron-beam structured-light technology instead of laser-structured light. However, it must be scanned during the deposition interval and it is impossible to calculate the height of the deposited layer in real time during the deposition process. So the real-time control of the droplet transfer cannot be realized. Taminger et al. [7] proposed a closed-loop EBF3 control method where a side-view optical camera can be used to monitor the height of a deposited bead on substrate. The Z-height of the deposit was adjusted up or down to maintain a consistent deposition distance and eliminate wire sticks and drips associated with incorrect standoff distance between the wire feeder and the deposit. However, this patent only gives the original idea, instead of the specific image-processing algorithm, as well as the realized detection accuracy and control accuracy.

In this paper, a vision-based method is proposed to monitor the droplet transfer distance and adjust the substrate height in real time to achieve a stable liquid bridge transfer.

## 2. Effect of Droplet Transfer Distance on Deposition Quality

### 2.1. Definition of Droplet Transfer Distance

A schematic of a typical droplet transfer is shown in Figure 1. The edge line of the electron beam near the side of the wire is denoted as L_0_. The long axis of the molten pool is denoted as L_1_. The central axis of the wire is denoted as L_2_. The intersection of L_0_ and L_1_ is denoted as P_1._ The intersection of L_0_ and L_2_ is denoted as P_2_. The distance between P_1_ and P_2_ is defined as △h, which represents the droplet transfer distance.

### 2.2. Different Droplet Transfer States Under Different Droplet Transfer Distances

A 50 mm line deposit is produced with the droplet transfer distance changed linearly from 0 mm to 5 mm. The experimental parameters are shown in Table 1.

The droplet transfer states under different droplet transfer distances are shown in Figure 2. It can be seen that in 0~1 mm, the droplet transfer presents a liquid bridge transfer form. When the droplet transfer distance is 1~5 mm, large droplet transfer begins to appear, and the droplet diameter gradually increases.

## 3. Image Processing Algorithm for Calculation of Droplet Transfer Distance

As mentioned in the previous section, in order to obtain the droplet transfer distance, it is necessary to obtain the position of the intersection point P_1_ between the long axis of the molten pool and the sideline of the electron beam, in order to calculate the distance between P_1_ and P_2_. The drop transfer images obtained by a camera usually have noise points; at the same time, there will be interference factors such as big droplet occlusion and miscellaneous-light reflection in the process of deposition. Hence, the methods of image processing are required to obtain the droplet transfer distance quickly and stably. The entire image processing flow proposed in this study is shown in Figure 3a.

### 3.1. Image Preprocessing

The droplet transfer image is captured by an industrial camera, as shown in Figure 4a. Firstly, the original image is binarized on a preset threshold to facilitate the extraction of the molten pool. The binarized droplet transfer image is shown in Figure 4b. It can be seen from Figure 4b that after being binarized, some miscellaneous points and solidification areas around the lower grey value have been eliminated. There are still some high-level noise points in the figure due to spatter and reflection. The image is first eroded and then dilated by a square connected domain with a size of 20. The droplet transfer image after erosion/dilatation is shown in Figure 4c. As we can see from the figure, the spatter and small reflection area can be eliminated.

### 3.2. Image Segmentation

The preprocessed images are divided into two regions: the droplet and the molten pool overlapping regions and the reflection regions. In order to accurately calculate the long axis of the molten pool, it is necessary to segment the image and accurately separate the droplet, the molten pool and reflection area. The flow chart is shown in Figure 3b,c. The image is accumulated along the row according to Equation (1), and the pixel accumulation distribution curve (Y*_i_*) along the image row is obtained, as shown in Figure 5a. First, the maximum peak point of the whole curve is found, and the row index (*i_max_*) of this point is regarded as the height value at the maximum length of the molten pool. Then, it is necessary to find the local minimum point of the curve (Y*_i_*) from *i* = (*i_max_* -200) to *i* = *i_max_*. The row index (*i_droplet_*) of this point is considered the height of the horizontal boundary between the molten pool and the droplet. Afterward, it is necessary to find the local minimum point of the curve (Y*_i_*) from *i* = *i_max_* to *i* = (*i_max_* +200). The row index (*i_reflection_*) of this point is considered the height of the horizontal boundary between the molten pool and reflection. The segmented image is shown in Figure 5b. The aspect ratio and center of the smallest external rectangle of each region are then calculated, respectively. In the connected domain where the aspect ratio is greater than 4 and the pixel area is greater than 5000, the closest connected domain to the height at the maximum length of the molten pool is considered as the molten pool.
(1)$$Y_{i}=\sum_{j=1}^{j=1168}I\left(i,j\right)$$
where *i* is the row index of the image, *j* is the column index of the image, *I*(*i,j*) is the grey value of the pixel in row i, column j of the image, *Y_i_* is the cumulative grey value in row *i* of the image.

### 3.3. Calculation of Droplet Transfer Distance

After obtaining the molten pool area, it is necessary to extract the long axis L_1_ of the molten pool in Figure 6a, whose flow chart is shown in Figure 3d. The left and right limit points T_1_ and T_2_ of the molten pool are first extracted, as shown in Figure 6b. To avoid unnecessary fluctuation of the molten pool majoraxis detection caused by the drift of the marginal point, the left and right limit points are moved to 10% of the maximum molten pool length inside the molten pool. After that, the midpoint M_1_ and M_2_ of the upper and lower points of the pool profile is calculated at this horizontal position. The line of M_1_ and M_2_ was used as the molten pool long axis L_1_. Because the position of the wire and electron beam is fixed, the position of them is calibrated in advance of the experiment. Taking the intersection point P_1_ of L_1_ and L_0_, and the intersection point P_2_ of L_2_ and L_0_, the pixel distance of P_1_ and P_2_ is calculated as △h’, denoting the droplet transfer distance in pixels.

### 3.4. Transformation From Pixel to Millimeter

The camera should be calibrated to calculate the distance of the droplet transfer △h in millimeters from the distance △h’ in pixels. The principle of calibration is shown in Figure 7. After the electron beam is emitted from the cathode of the electron gun, it is vertically incident on the substrate surface, intersecting with the deposition surface at point P_1_, and intersecting with the wire at point P_2_. P_1_ passes through the lens and focuses on point B_1_ of the CMOS chip. The real distance of the droplet transfer is defined as △h, and the pixel distance of the droplet transfer is defined as △h'. The angle between the electron beam and the imaging light axis is defined as α. The focal length of lens is f. The object distance is u1 and the image distance is v1. The corresponding relation between the pixel distance △h’ of the droplet transfer and the real distance △h of the droplet transfer can then be obtained by geometric calculation:(2)$$△h=\frac{u1}{v1*\sin\alpha }*△h'$$

## 4. Experiment of Closed-Loop Control of Droplet Transfer

### 4.1. Introduction of Experimental Equipment

The entire system is schematically shown in Figure 8. The camera used to capture the droplet transfer image sends the image to the industrial control computer through the network cable. The industrial control computer obtains the droplet transfer distance △h by image processing. The deviation e is obtained by subtracting the detected droplet transfer distance △h from the preset-expected droplet transfer distance △h_0_. In order to realize the closed-loop control of droplet transfer distance, the droplet transfer deviation e is sent to the numeric platform by the OPC-UA communication module of the Siemens 840Dsl CNC system. The 840Dsl CNC system then compensates the droplet transfer distance in real time by adjusting the platform height according to e. The physical system is shown in Figure 9. The system consists of an 15 kW electron gun for generating an electron beam. The maximum acceleration voltage is 60 kV. A three-degrees-of-freedom motion platform for placing the substrate is located in the vacuum chamber. The motion range of the platform is 1000 mm (X) × 500 mm (Y) × 500 mm (Z). An Image Source industrial camera is adopted to acquire the droplet transfer image. The frame rate is 20 fps. The CMOS sensor chip size is 1 inch, with a resolution of 4096*2160 (the pixel after ROI is 1292*1168). The pixel size is 3.45 μm × 3.45 μm. The industrial control computer is configured with an E5-1650 processor and 32GB memory.

### 4.2. Flowchart of the Closed-Loop Control System

The flowchart of the entire system is shown in Figure 10. The electronic gun and the motion platform are controlled by the Siemens 840Dsl numerical control system. The electron beam is accelerated by a high-voltage electric field to melt the wire and substrate to form a droplet transfer. At this point, the IPC generates a signal that triggers the camera to capture the droplet transfer image. The image collected by the camera is transmitted to the industrial control computer through the Gig-e port. The droplet transfer pixel distance is obtained by the image processing program. The real distance of the droplet transfer is calculated according to Equation (2) in the previous section. The droplet transfer distance deviation e is calculated and sent to the controller. The output value of the controller is transmitted to the 840dsl NC system through the OPCUA communication. Then, the height of the platform is adjusted in real time to make the droplet transfer distance stable.

## 5. Results

### 5.1. Detection Accuracy of Droplet Transfer Distance

For the open-loop experiment described in Table 1, the droplet transfer distance is detected by the image processing algorithm. Figure 11 shows the detected droplet transfer distance versus the actual droplet transfer distance. It can be seen from the figure that the linearity of the detection method is very high and the correlation coefficient reaches up to 0.99. Figure 12 is the measurement error versus the actual droplet transfer distance. Ninety-five percent of the points fall within ± 0.08 mm around the actual droplet transfer distance. The maximum error is 0.14 mm, which meets the demand of the closed-loop control of droplet transfer.

### 5.2. Closed-Loop Control Accuracy of Droplet Transfer Distance

The real-time droplet transfer distance within the closed-loop control is shown in Figure 13. It can be seen that the liquid bridge length can be maintained within ±0.2 mm uniformly and stably.

The deposition effect of the open-loop and closed-loop control is shown in Figure 14. It can be seen that, when the droplet transfer distance changes from 0 to 5 mm without control, the droplet transfer state changes from liquid bridge to large droplet transfer, and the deposition quality is poor. When the droplet transfer state is maintained as a liquid bridge transfer within closed-loop control of the droplet transfer distance, the deposition quality is consistent, which is better than that without control.

## 6. Conclusions

In this paper, a droplet transfer detection system is outlined which can clearly photograph the droplet transfer image and the stable liquid bridge process interval (0~1 mm) is obtained by different droplet transfer distance experiments. In addition, a method of droplet transfer distance detection is proposed, and the detection accuracy is ± 0.1 mm. Moreover, a closed-loop control method for droplet transfer in EBF3 based on image processing is proposed. In the closed-loop control experiment, the droplet transfer can be realized stably. Furthermore, the control accuracy is ± 0.2 mm. The deposition quality is effectively improved using this closed-loop control method, which provides a basis for future metal additive manufacturing in space. In addition, a custom-built image processing algorithm is developed in order to improve the detection stability of droplet transfer. The reliability of the algorithm is improved by threshold binarization, local minimum segmentation, and dilatation/erosion.

## Figures and Tables

**Figure 1 sensors-20-00923-f001:**
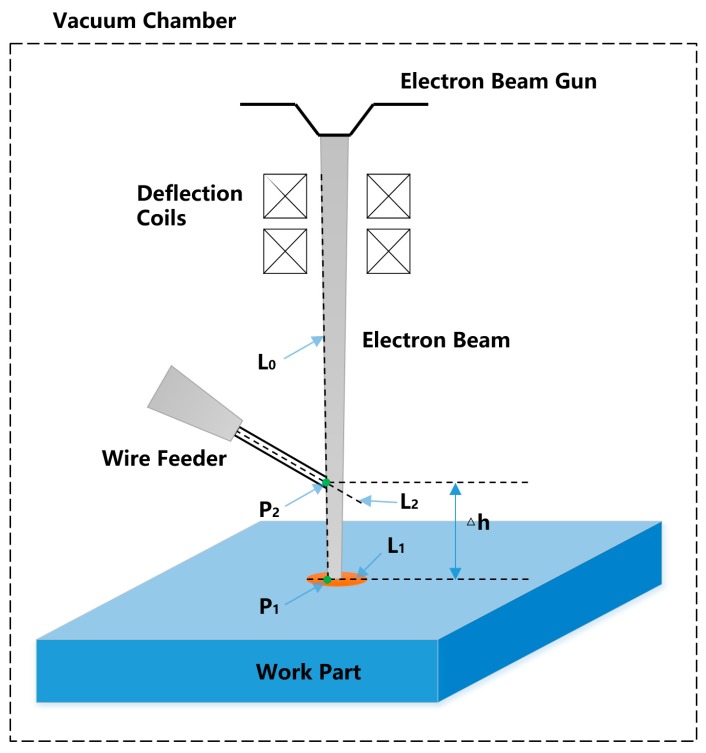
Definition of droplet transfer distance.

**Figure 2 sensors-20-00923-f002:**
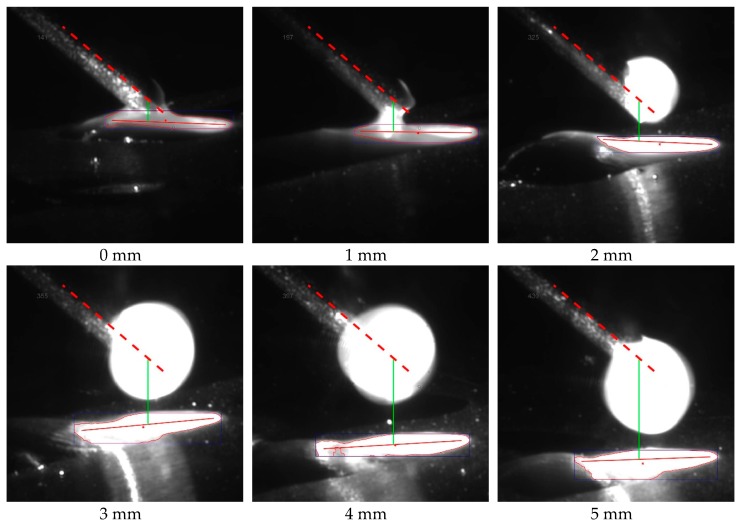
Different droplet transfer states under different droplet transfer distances.

**Figure 3 sensors-20-00923-f003:**
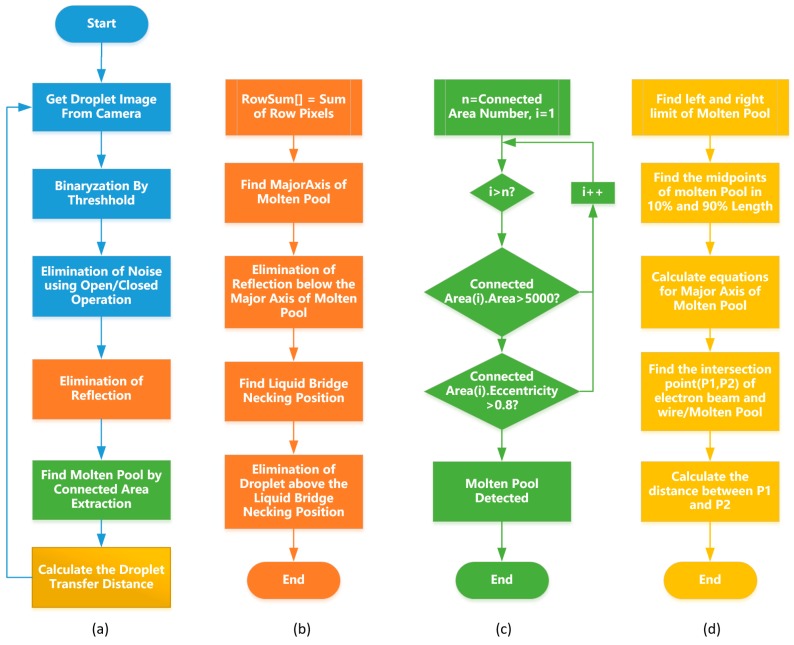
Flowchart of image processing: (**a**) main program of image processing algorithm; (**b**) subroutine for image segmentation; (**c**) subroutine for molten-pool extraction; (**d**) subroutine for calculation of droplet transfer distance.

**Figure 4 sensors-20-00923-f004:**
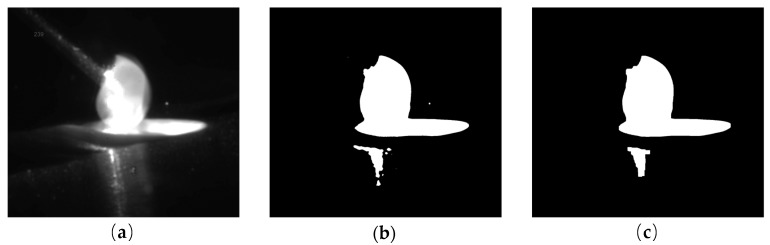
Image sequence of image preprocessing: (**a**) original droplet transfer image; (**b**) image after binarization; (**c**) image after erosion/dilatation.

**Figure 5 sensors-20-00923-f005:**
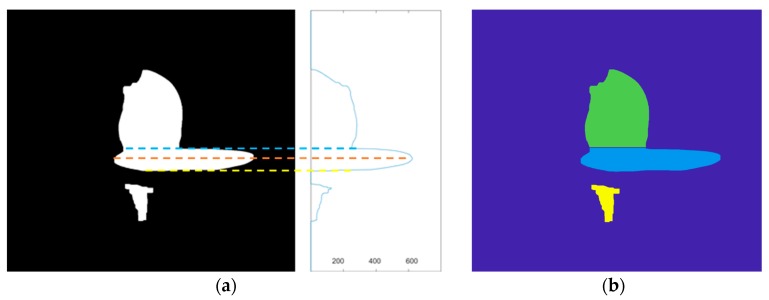
(**a**) Accumulation along the image row, where the orange dotted lines indicate the height of the maximum length of the molten pool, the blue dotted lines indicate the height of the boundary between the molten pool and droplet, the yellow dotted lines indicate the height of the boundary between the molten pool and reflection. The horizontal axis represents Yi (the cumulative grey value in row i of the image). (**b**) Image segmentation result.

**Figure 6 sensors-20-00923-f006:**
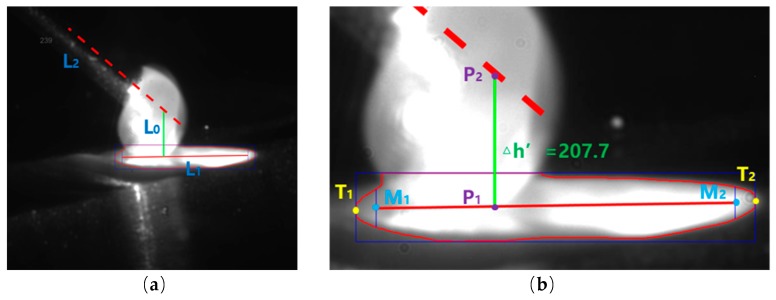
(**a**) Advance calibration of L_0_ and L_2_; (**b**) calculation of droplet transfer distance in pixels.

**Figure 7 sensors-20-00923-f007:**
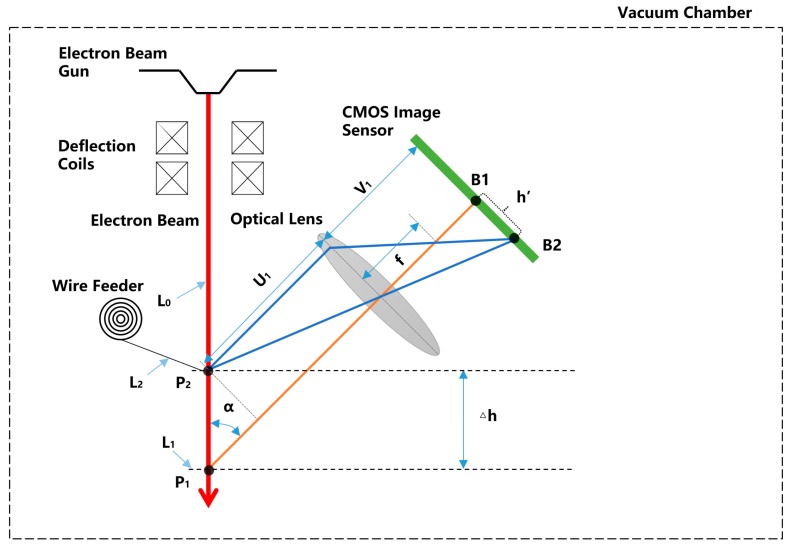
Principle of transformation from pixels to millimeters.

**Figure 8 sensors-20-00923-f008:**
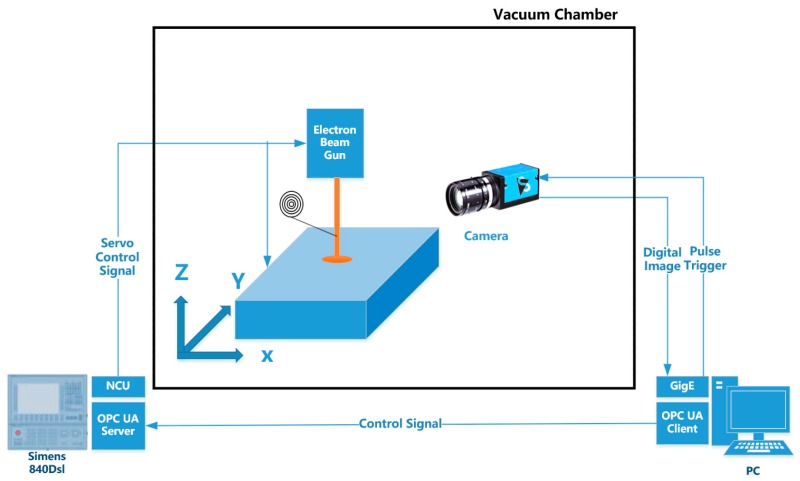
Composition of the closed-loop control system.

**Figure 9 sensors-20-00923-f009:**
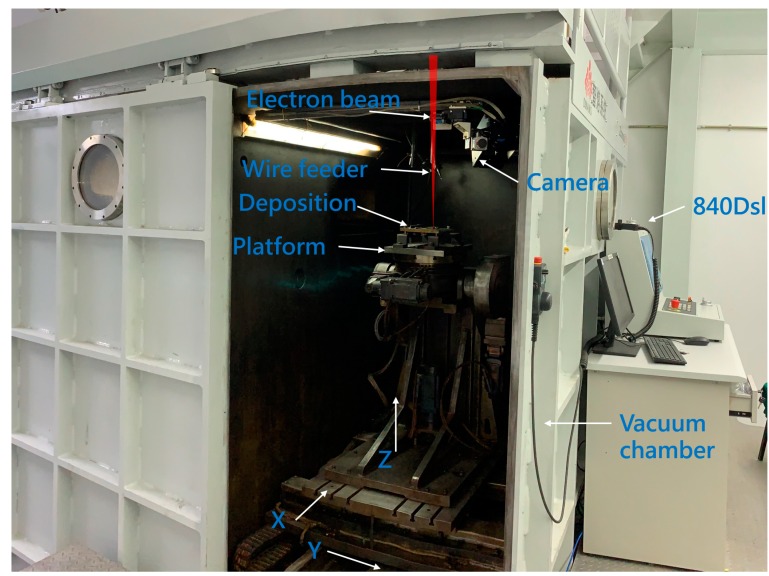
Physical system for the closed-loop control experiment.

**Figure 10 sensors-20-00923-f010:**
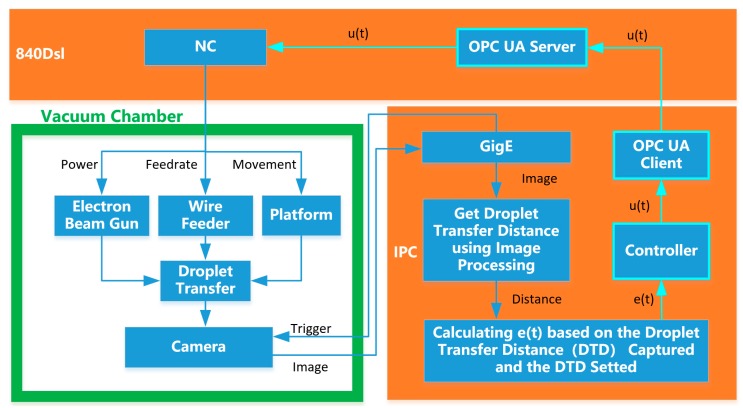
Flow chart of the measurement system.

**Figure 11 sensors-20-00923-f011:**
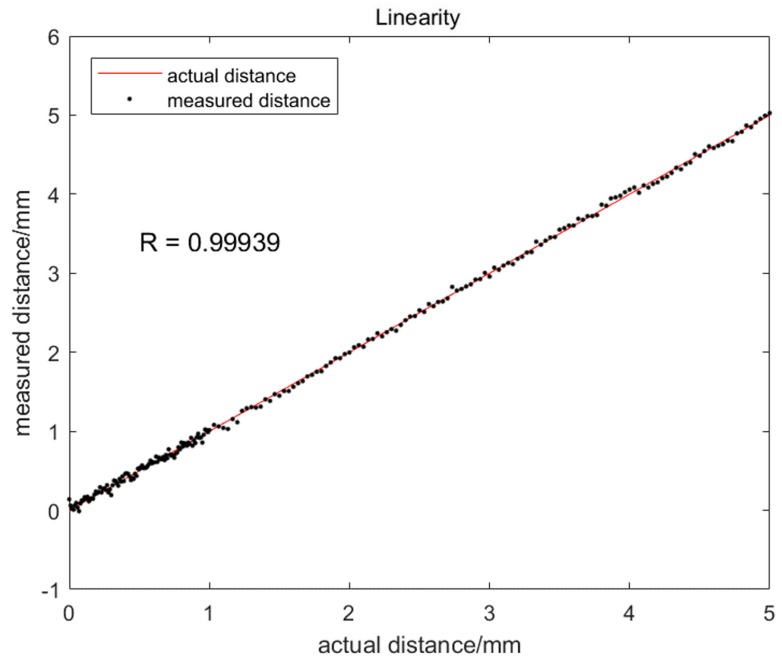
Linearity of Droplet Transfer Distance Detection Method.

**Figure 12 sensors-20-00923-f012:**
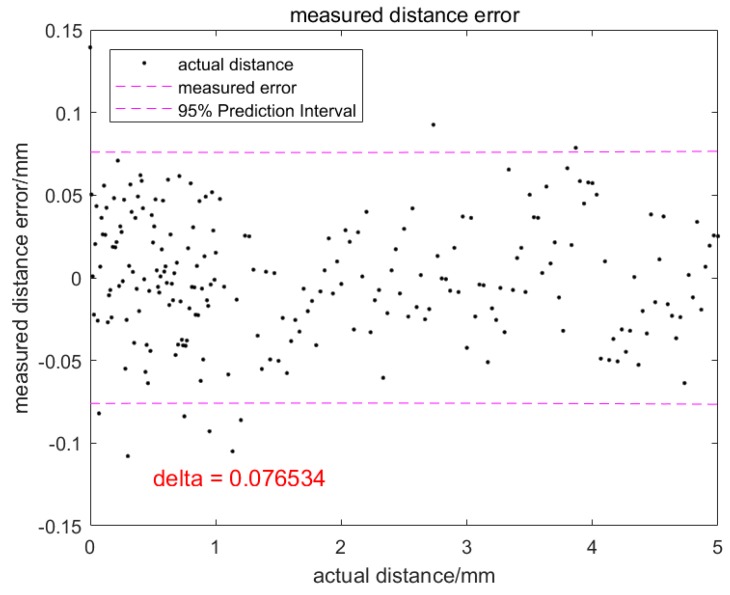
Droplet transfer distance detection error.

**Figure 13 sensors-20-00923-f013:**
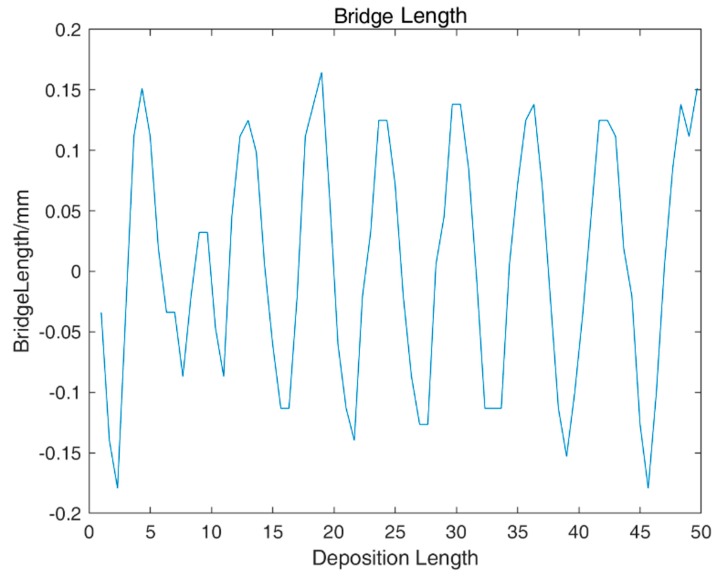
Actual liquid bridge length during closed-loop control of the droplet transfer.

**Figure 14 sensors-20-00923-f014:**
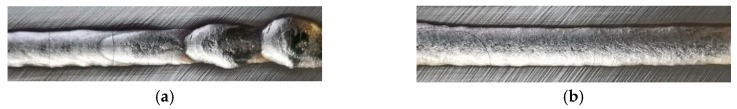
Deposition result: (**a**) deposition within the open-loop control; (**b**) deposition within the closed-loop control.

**Table 1 sensors-20-00923-t001:** Experimental parameters.

Parameter	Value	Unit
Acceleration voltage	50	kV
Electron-beam current	50	mA
Deposition speed	400	mm/min
Wire-feed speed	1.5	m/min
Droplet transfer distance	0~5	mm
Deposition length	50	mm

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
