# Peer review of "Closed-Loop Control of Droplet Transfer in Electron-Beam Freeform Fabrication"

_sensors, 2020, doi:10.3390/s20030923_

Round 1

Reviewer 1 Report

Comments:

In the article “Close Loop Control of Droplet Transfer in Electron Beam Freeform Fabrication” authors report real-time and close-loop control of the droplet transfer method based on machine vision and realize the detection accuracy of this proposed system upto +/- 0.08 mm.

I suggested change as closed-loop control instead of close loop control" The main concern in this paper author not defined problem statement properly for this work in the abstract as well as in the introduction. The author should clarify what is different between this work and already published patterned (Patent No.: US 8,452,073 B2 , date: May 28, 2013). The author should explain the problem of how much accuracy achieved so far and how much improved in this work. In the abstract, the author reported this system to stabilize the droplet transfer distance within +/- 0.15 mm but in the result mentioned +/- 0.14 mm, the author should consistency with their results. The author should describe the eq.1 (what is mean I, i, j, etc.,). The author should rewrite the following sentence in section 3.2. image segmentation “Find the local minimum Yi within 200 pixels above the maximum length of the molten pool, which is considered as the boundary between molten pool and droplet. Find the local minimum Yi within 200 pixels below the maximum length of the molten pool, which is considered as the boundary between the molten pool and reflection.” It is hard to get the point across, author mentions their message simple and straight to the point.   In Fig.5, the author should give what are the dotted lines mean in the figure and what are the number 200, 400, 600 etc., Section 3.4 should come before 3.3 The author should give what is the unit of the number mentioned in the figure 6. “ 2707.7” Check the spelling Figure 11 “linierty”

Based on such arguments I cannot recommend its publication in the sensor at this current form of the manuscript.

Reviewer 2 Report

The article carried out important work to improve the quality of 3D printing by droplet transfer control. The developed control technology and correctly selected parameters can significantly improve the quality of the deposed material. The results obtained are of interest to a wide range of technology developers and materials scientists in the field of additive technologies. 

This paper is new, original and well organized.  English language is good in the paper and all references are adequate. Also all parts of paper are important and conclusion is fine. I recommend this paper for publication, but with minor revision:

1. For me, as a specialist in materials science, the phrase "uneven surface" sounds bad. In my opinion, it is better to replace this phrase with "rough surface".

2. The quality of Figure 8 should be improved. 

Reviewer 3 Report

The present paper discusses the real time control of the droplet transfer during EBF3.

The aim of the article is not clear. Therefore, the authors should clearly indicate the goal of this study. The paper is having only experimental approach without the real/proper problem definition in the field citing extensive references for the same. The authors must give a good introduction for the problem being followed and the already done work by others in the field by proper and extensive citation.

The experimental methodology must be validated to claim the results. The experiments are not enough (only one experiment is presented and discussed). More experiments with different technological parameters for EBF3 process must be investigated.

Round 2

Reviewer 1 Report

now looks better, I accept this manuscript at the present form.  

Reviewer 3 Report

The manuscript can be published in this version.